# Effect of Superhydrophobic Coating and Nanofiller Loading on Facial Elastomer Physical Properties

**DOI:** 10.3390/ma15207343

**Published:** 2022-10-20

**Authors:** Rahmi Khairani Aulia, Mark W. Beatty, Bobby Simetich

**Affiliations:** 1College of Dentistry, University of Nebraska Medical Center, Lincoln, NE 68583-0740, USA; 2Faculty of Dentistry, Andalas University, Padang 25175, West Sumatera, Indonesia; 3Veteran Affairs Nebraska-Western Iowa Healthcare System, Omaha, NE 68105-1850, USA

**Keywords:** facial prosthetic, physical properties, polydimethyl siloxane, nanoparticles

## Abstract

Facial prosthetics are currently constructed of materials that are far from optimal; superior materials with a “skin-like” feel are required. In this study, the property changes brought about by the consecutive additions of hydrophobic- and uncoated nano-SiO_2_ to polydimethylsiloxane (PDMS) are assessed, and the alterations are compared with those observed for conventional submicron SiO_2_-filled materials. In sequence, 0%, 0.5%, 5%, 10%, and 15% by weight of each filler type were successively added to vinyl-terminated PDMS. Tensile, tear, Durometer hardness, translucency, and viscoelastic properties were assessed, and hardness and translucency were further measured after 3000 h of outdoor weathering. The results showed that 15% coated nano- SiO_2_-filled PDMS materials given the highest tensile strength, elastic modulus, storage modulus, loss modulus, tear strength, and durometer hardness (*p* < 0.05), whereas 15% submicron coated SiO_2_-filled materials displayed the highest failure strain and translucency parameter (*p* < 0.05). Only 10%- and 15%-filled submicron SiO_2_ PDMS materials were altered by outdoor weathering; nevertheless, the increases were assessed to be too small to be clinically perceptible. As increased filler levels provided protection against solar radiation, heat, and moisture, only unfilled and 0.5%-filled PDMS formulations discolored from weathering. 15%-filled superhydrophobic-coated nano- SiO_2_-filled PDMS was found to produce the strongest, most tear-resistant, and least translucent materials, but it also produced materials with limited stretchability and high hardness, which were regarded to be downsides for creating a “skin-like” feel.

## 1. Introduction

Facial disfigurement is a distressing condition that leads to overwhelming psychological suffering because the defects are highly visible. This causes social isolation, anxiety and depression [1]. Reconstruction of significant facial defects is still a major challenge for the reconstructive facial plastic surgeon. Surgery often involves long and complicated procedures, to which many patients are ill-suited [2]. This situation makes the use of maxillofacial prosthetics critical to restoring facial disfigurement, both aesthetically and functionally [3]. Aesthetics is an essential factor in the acceptance of a facial prosthesis. The ideal material should resemble and feel like natural facial tissue and should retain these qualities throughout the prosthesis’ lifetime [4].

The most widely used materials for maxillofacial prostheses are silicone elastomers [5]. Due to their chemical inertness, adequate strength, longevity, easy manipulation, and biocompatibility, silicones are preferred [6]. Silicone has been proposed as the best material to rehabilitate patients with craniofacial defects because its flexibility delivers better comfort to the patient [7,8]. Filled polydimethylsiloxane (PDMS), a silicone, is the material most frequently utilized to make facial prosthetics [9]. Their two major components, the PDMS polymer and the silica fillers, have a significant impact on the quality of these materials. These components and their interactions have an impact on the prosthesis’s overall toughness, usable lifespan, and biomimetic texture [6,10]. However, the surface texture is a commonly reported problem, along with discoloration, lack of longevity, material degradation, margin deterioration, and decreased mechanical adequacy. The current generation of maxillofacial prosthetic materials is perceived as too stiff and not mimicking the natural feel of skin [11].

Results from tensile and tear tests are commonly reported for facial prosthetic materials because they reflect a material’s ability to stretch until fracture ensues. This is particularly important, as a prosthesis is vulnerable to failure along thin edges during its removal from the face, especially if an adhesive is used for retention. Ideally the material should exhibit good stretchability without fracture when pulled away from facial tissues. [6,10]. Measurements of tensile strength, modulus and breaking strain provide quantitative assessments of a material’s stretchability and susceptibility to tensile failure under laboratory conditions. Although clinical performance cannot be accurately predicted by these tests, they permit material comparisons and enhance the understanding of a material’s behavior when challenged in this manner. If a flaw develops within a prosthesis, then the force to produce fracture decreases, and continues to decrease as the flaw elongates [6]. Results from tear strength measurements provide insight into fracture resistance under these circumstances.

Prosthesis appearance and feel during daily wear is crucial to patient satisfaction [12]. Pigments, fillers and polymer contribute to these qualities, and changes to any component can produce either beneficial or detrimental results. When polymer-filler ratios are changed, translucency, compressibility and damping behavior may be reduced, thereby rendering a prosthesis lifeless in its appearance, feel and ability to move in concert with underlying tissues. Tests for translucency, Durometer hardness and dynamic mechanical analysis are typically chosen to identify these changes and better understand interactions occurring among various elastomer components [10].

There is continuing research to create novel polymeric materials that feel more “skin-like” and have enhanced physical properties. Recent research has focused on incorporating nanoparticle fillers into the organic polymer matrix, creating a nanocomposite that better combines the strength of the filler with flexibility of the organic matrix [12]. Previous research reported that the addition of 5% by weight nano-SiO_2_ to polydimethylsiloxane produced elastomers with good stretchability and strength while maintaining low stiffness. These conditions were deemed essential to achieving a proper texture and “feel” in a maxillofacial prosthesis [13]. Other research reported increases in tensile strength, tear strength, and percentage elongation when 3% by weight nano-SiO_2_ was added to commercial PDMS materials containing conventional submicron size fillers. This is expected to permit construction of a prosthesis with a thinner margin that offers greater stretching and a lower chance of tearing [14].

Evenly dispersed nanoparticles are key to achieving uniform material properties. Therefore, recent attention has been given to a class of hydrophobic coatings that are sometimes classified as “superhydrophobic” as means to improve dispersion and interaction with polymer molecules. Superhydrophobic surfaces exhibit extremely high-water repellency, where upon being dropped from a height onto the surface, water droplets bead up, roll with a minimal force, and bounce [15]. The water contact angle must be larger than 120 or 150 degrees for a surface to be classified as superhydrophobic [16,17]. Dimethoxydiphenyl silane (DMDPS, or KH220) contact angle has been reported to be 133 degrees, which falls into the range of superhydrophobicity [18]. Coatings serving as “superhydrophobic” coatings are expected to invoke repulsion among particles and thereby assist in separating nanoparticles during their incorporation into polymer. In principle this should produce a more uniformly distributed filler phase. The purpose of this research is to study PDMS-filled with uncoated and hydrophobic-coated nano-silica (nano-SiO_2_) and compare their physical behaviours with those measured for a conventional submicron-filled PDMS.

Given this background information, we hypothesize that sequential additions of superhydrophobic-coated nano-SiO_2_ to PDMS will generate improved physical properties than observed with PDMS filled with either uncoated nano-SiO_2_ or conventional submicron-filled silica.

## 2. Materials and Methods

### 2.1. Preparation of Samples

Elastomers were constructed by combining vinyl-terminated PDMS, SiO_2_ fillers, crosslinker and platinum catalyst, then polymerizing the mixture under heat. Three types of SiO_2_ fillers, 15 nm coated nano-SiO_2_ (KH220), 15 nm uncoated nano-SiO_2_ and 200 nm submicron SiO_2_ (TS530) were used for this project (Table 1). Nano-SiO_2_ were chosen based on previous research showing that the addition of 5% nano-SiO_2_ to PDMS produced high strength elastomer and good ability to stretch while maintaining low stiffness [13]. Submicron SiO_2_ was chosen as a control because it serves as a conventional filler in PDMS.

For elastomer preparation, each type of filler was sequentially added to vinyl terminated PDMS at 0%, 0.5%, 5%, 10%, and 15% by weight. A procedural flow chart is presented in Figure 1. The filler particles were first incorporated using a rotary mixer (ME 100L, Charles Ross and Son Company, Hauppauge, NY, USA) at 5000 rpm for ten minutes. For nanoparticles (KH220 and Uncoated), an ultrasonic processor (Hielscher model UP200S, Teltow, Germany) was applied for ten minutes at 105 W/cm^2^ to burst nanoparticle agglomerates. Each mixture was contained in a 100 mL stainless-steel cup that was cooled in an ice bath over a ten-minute period. To ensure that the nanoparticles were distributed evenly after being removed from the ultrasonic mixer, the mixture was rotary mixed once more with a Cowles disperser for 10 min at 5000 rpm. For traditional submicron particles (TS530), rotary dispersion was accomplished without ultrasonic treatment.

For polymerization, the filler-containing vinyl-terminated PDMS was combined in a 1:1 molar ratio with polymethyl hydrogen siloxane (V-XL crosslinker) and 10 ppm platinum catalyst. The mixtures were hand-mixed in a paper cup using a wooden tongue depressor for about two minutes, then placed into a high vacuum pump (8920 Vacuum Pump DirecTorr, Welch Vacuum Technology, Skokie, IL, USA) under 5 × 10^−3^ torr constant vacuum. Bubble removal was visually checked, usually requiring five to ten minutes. The bubble-free mixture then was slowly poured into a mold, and a lid was placed with sufficient pressure to squeeze out excess material. The pressed mold was placed into an 85 °C oven for sixty minutes for polymerization.

Two types of molds were used in this project. For tensile properties, elastomer mixtures were poured onto a rectangular gypsum mold, covered and tightened with six clamps to extrude excess materials. This mold produced 245 mm length × 165 mm width × 2 mm thickness elastomer sheets. Dumbbell-shaped samples were cut from elastomer sheets using ASTM D412 die C die cutter to produce twelve samples for each group [19]. For tear strength, the same mold was used to make the elastomer sheets and ASTM D624 die C die cutter was used to construct ten samples [19]. Test sample dimensions for ASTM tests are presented in Figure 2. For durometer hardness and translucency parameter, molds were assembled from polyvinyl chloride (PVC) pipes cut into 6 mm thick rings. Each ring was placed onto a gypsum block and held in place by injecting medium-body polyvinyl siloxane (PVS) impression material around the periphery. The siloxane mixture then was poured slowly into the mold and a glass lab placed on top to squeeze out excess material. This mold created a 33 mm diameter × 6 mm thickness disc, as compliant with ASTM D2240 [20]. Each of the filler loading groups in three different types of filler were divided into two subgroups, one for indoor storage and the other exposed to outdoor weathering with five discs per subgroup [19].

### 2.2. Physical Properties Measurements

#### 2.2.1. Tensile Properties Measurements

A universal testing machine (Instron 1123-5500R, Instron Corp., Boston, MA, USA) controlled by Bluehill 3 software were used to perform tensile testing and record tensile strength, elastic modulus and failure strain data. Each sample was loaded into pneumatic grips with 30 psi pressure and an extensometer was placed with a gauge length of 25 mm. The dumbbell was elongated at 500 mm/min rate while the computer recorded stress versus strain data until failure. Electronic measurements of force and elongation were taken, and these equations were utilized to create the stress–strain tensile curves [4].
$$\sigma =\frac{F}{A}\;\;\;\;\;\varepsilon =\frac{\Delta L}{L_{0}}$$
where *σ* is stress (MPa) and *ɛ* is strain (mm/mm). *F* and Δ*L*, are the electronically collected force (N) and deformation (mm), respectively, while *A* is the cross-section area (mm^2^) and *L*_0_ is the initial (gauge) length of the samples (mm).

#### 2.2.2. Tear Strength Measurement

The same universal testing machine and software used for tensile properties also performed tear strength measurements. The test piece was measured for thickness and width, then loaded into pneumatic grips with 30 psi pressure. The rate of jaw separation was 500 mm/min and the maximum force was recorded until rupture. Tear strength is calculated by dividing the highest force needed to break the sample by its thickness and calculated using the following equation [10]:$$T_{s}=\frac{F}{t}$$
where *T*_s_ is the tear strength (N/mm), *F* the load at failure (N), *t* the thickness of specimen (mm).

#### 2.2.3. Shore a Hardness Measurement

The previously described sample discs were used for durometer hardness (10 per group). Each of the filler loading groups in three different types of filler were divided into two subgroups, one for indoor storage (n = 5) and the other exposed to outdoor weathering (n = 5). Indoor samples were placed inside a closed box held within a controlled temperature and humidity darkroom at the Biomaterials Laboratory, College of Dentistry, University of Nebraska Medical Center. Outdoor samples were placed on the roof of Keim Hall located at the East Campus University of Nebraska-Lincoln. Weathering was performed for 3000 h which represent approximately 3 years of prothesis clinical year (approximately three hours of sun exposures per day). Weathering exposures commenced 1 July 2019, and ended 5 November 2019. Hardness measurements were taken before and after weathering with a shore A hardness tester mounted on a loading stand. Five measurements were made at five distinct locations on a disc’s surface in accordance with the ASTM D2240 standard, and the average of those readings was used to calculate the hardness of that particular disc.

#### 2.2.4. Translucency Parameter Measurement

A color reflectance spectrophotometer (CM-2002, Konica Minolta Corp., Ramsey, NJ, USA) with computer software (SpectraMagic NX, Konica Minolta Corp., Ramsey, NJ, USA), daylight illuminant (D65), and a viewing angle of 10° was used for measurement. Color coordinates were measured in CIElab color space. All measurements were performed before and after outdoor weathering, with measurements performed on the same surfaces before and after weathering. Translucency parameter (TP) was calculated as the three-dimensional color difference vector between measurements obtained on white and black backgrounds as follows [21]
$$TP=[{\left(L_{B}^{*}-L_{W}^{*}\right)}^{2}+{\left(a_{B}^{*}-a_{w}^{*}\right)}^{2}+{\left(b_{B}^{*}-b_{W}^{*}\right)}^{2}{]}^{1/2}$$
where the *L** axis is white-black, the *a** axis is red-green, the *b** is the yellow-blue, the subscript *B* refers to the black backing color parameter, and the subscript *W* refers to the white backing color parameter.

#### 2.2.5. Viscoelastic Properties Measurement

Viscoelastic properties were measured using a Dynamic Mechanical Analyzer (DMA) (DMA 8000, Perkin Elmer, Inc., Waltham, MA, USA) in tensile mode. This test was performed to determine elastic and damping properties of the PDMS formulations. The experiments were carried out in tension at a frequency of 1.0 Hz with 0.02 mm displacement in the temperature range between −150 °C and 20 °C, at a heating rate of 2 °C/min. The temperature range was chosen based on preliminary tests that identified key thermal transitions. Rectangular samples 5 mm length × 6.3 mm width × 2.5 mm thickness were tested for storage modulus, loss modulus, and tan delta (n = 3 per group). The storage modulus determined the elastic behavior of the elastomer and the ratio of the loss modulus to the storage modulus (tan delta) was a measure of the energy dissipation of a material, or its damping [22].

### 2.3. Microscopic Analysis

To obtain the microscopic characterization and dispersion of the nanoparticle fillers, scanning electronic microscopy (SEM) was used to observe the bulk surface of the specimen. The samples were taken from approximately 1 mm thick elastomer sections for each of the three filler types. Samples were sputter coated with Cressington 108 (Watford, UK) that was configured to the following parameters: working distance (5 cm), material constant (Au, 0.9), sputtering current (40 mA), and Ar pressure (0.05 mbar). This resulted in coating thickness of about 1 nm. The images were taken using SEM (Helios NanoLab 660, Thermo Fisher Scientific, Waltham, MA, USA) at 500 times and 50,000 times magnification, with 2.0 kV acceleration voltage.

### 2.4. Data Analysis

For tensile properties, group means and standard errors were calculated for dependent variables ultimate tensile strength, elastic modulus, and failure strain, while independent variables were filler type (KH220, Uncoated and TS530) and filler loading (0%, 0.5%, 5%, 10%, and 15%). Each of the groups consisted of 12 samples, therefore, total samples that used in this test were 12 samples/group × 3 filler types × 5 filler loading = 180 samples. The null hypothesis was that tensile properties were not affected by filler type and filler loadings. This was tested by two-way ANOVA with Tukey post hoc test at *p* < 0.05 level of confidence.

For tear strength, group means and standard errors were calculated for the dependent variable tear strength, while independent variables were filler type (KH220, Uncoated and TS530) and filler loading (0%, 0.5%, 5%, 10%, and 15%). Each of the groups consisted of 10 samples, therefore, total sample numbers were 10 samples × 3 filler types × 5 filler loading = 150 samples. The null hypothesis tested was that tear strength was not affected by filler type and filler loadings. This was tested by two-way ANOVA with Tukey post hoc test at *p* < 0.05 level of confidence.

For durometer hardness and translucency parameter, group means and standard errors were calculated for dependent variables (hardness and translucency parameter). Independent variables were filler type (KH220, Uncoated and TS530), filler loading (0, 0.5, 5, 10, and 15%), weathering exposures (indoor and outdoor), and weathering time (before and after 3000 h). Each of the groups consisted of 10 samples, then these samples were divided into 5 samples per indoor and outdoor group, with total samples tested being 150. The null hypothesis tested was that hardness and translucency parameters were not affected by filler type, filler loading, weathering exposure, and time. This was tested by four-way ANOVA with Tukey post hoc test at *p* < 0.05 level of confidence.

For viscoelastic properties, group means and standard errors were calculated for dependent variables (storage modulus, loss modulus, and tan delta), while independent variables were filler type (KH220, Uncoated and TS530), filler loading (0%, 0.5%, 5%, 10%, and 15%), and transitions (transition I, II, and III). The null hypothesis tested was that viscoelastic properties was not affected by filler type and filler loadings. This was tested by two-way ANOVA with Tukey post hoc test at *p* < 0.05 level of confidence.

## 3. Results

### 3.1. Nanoparticle Dispersion

The dispersion of nanoparticles at 15% filler loadings was assessed from Scanning Electron Microscopy (SEM) micrographs. This filler loading was chosen because it was expected to be the most difficult to disperse and would permit the easiest SEM observation. The dispersion of KH220 filler particles is shown in Figure 3a,b with 500× and 50,000× magnification, respectively. From Figure 3a, the images show that the filler particles are dispersed evenly in PDMS and with few agglomerates. A more detailed view of particle size and distribution can be observed at higher magnification in Figure 3b, where approximately 15 to 50 nm particle diameters are evident.

Uncoated filler particles showed uneven dispersion compared to KH220 particles (Figure 3c,d). In Figure 3c, at 500× magnification, certain particles appear as 1 to 10 μm-sized diameter clumps, indicative of particle agglomeration. At 50,000× magnification, a 1.5 μm-sized aggregate appears to contain particles ranging from approximately 15 nm to 300 nm in diameter (Figure 3d).

Figure 3e,f show dispersion of TS530 submicron fillers. Filler particles are evenly distributed in polymer, as shown at 800× magnification in Figure 3e. However, fewer particle numbers are observed compared to KH220 and uncoated nanoparticles. Particle sizes estimated from the micron marker present in Figure 3f, demonstrate 150 to 200 nm size particles.

### 3.2. Physical Properties

#### 3.2.1. Tensile Properties

Representative stress–strain plots comparing materials at each filler level are presented in Figure 4, Figure 5, Figure 6 and Figure 7. For each material group, the material closest to group mean values for tensile strength, modulus and breaking strain was chosen to serve as the representative plot.

##### Tensile Strength

Plots of tensile strength versus percent weight filler for each group are presented in Figure 8. For all groups, the tensile strength increased non-linearly as filler content increased.

Tensile strength values among the three filler types were not significantly different at lower filler loadings (0% and 0.5%, *p* ≥ 0.05). At 5% loadings and above, KH220-filled materials were significantly stronger than those filled with TS530, and stronger than Uncoated at 15% filler loading (all *p* < 0.05). No significant differences in tensile strength were noted at any filler loading between uncoated- and TS530-filled materials (*p* ≥ 0.05). The highest tensile strength was recorded for 15% filled KH220 materials, which were 1.4 times higher than 15% uncoated-filled material and 1.3 times higher than 15% TS530-filled materials. Compared to unfilled PDMS, 15% KH220-, uncoated-, and TS530-filled materials were 6, 4.3, and 4.2 times stronger, respectively.

##### Elastic Modulus

Plots of elastic modulus versus weight percent filler loading are presented in Figure 9. Results presented in Figure 9 show non-linear elastic moduli increases for KH220- and uncoated-filled materials, whereas modulus values reached a plateau at 10 weight percent filler for TS530-filled elastomers. Mean elastic modulus values among the three types of filler were not significantly different at low filler loadings (0% and 0.5%, *p* ≥ 0.05), but above 5%, elastic modulus, KH220- and Uncoated-filled materials were significantly greater than TS530 materials (*p* < 0.05). No significant differences in modulus were observed between KH220- and uncoated-filled materials until filler loadings reached 10% and 15% (*p* < 0.05). TS530-filled materials maintained lower elastic modulus values throughout all filler loading increases. At 5% loading, TS530-filled samples produced 1.3 times lower elastic modulus than both KH220- and uncoated-filled samples and the gap widened at higher filler loadings. The highest mean elastic modulus was recorded for 15% KH220-filled materials, which were 1.5 times higher than uncoated-filled materials and 2.5 times higher than TS530-filled materials. KH220-, uncoated- and TS530-filled materials at 15% loading were 4.0, 3.5, and 1.7 times stiffer than unfilled PDMS, respectively.

##### Failure Strain

Plots of failure strain versus weight percent filler from pairwise comparisons are presented in Figure 10. Results presented in Figure 10 show failure strain changes for the three filler types as filler loading increases non-linearly. The trends were inverse compared to tensile strength and elastic modulus plots, where TS530-filled materials showed higher values than KH220- and uncoated-filled materials.

There were no significant differences of failure strain at lower filler loadings (0%, 0.5%, and 5%, *p* < 0.05) for the three filler types, except between 0.5% uncoated- and TS530-filled materials (*p* < 0.05). Failure strain of TS530-filled materials were significantly greater compared to KH220- and uncoated-filled materials for 10% and 15% filler loadings, while failure strain for 15% of KH220-filled materials were significantly greater than uncoated-filled materials (*p* < 0.05). The highest failure strain among all groups was for 15% TS530 filler, which was 1.4 times higher than 15% KH220-filled materials and 1.6 times higher than uncoated-filled materials. 15% filled KH220-, uncoated- and TS530-materials produced 2-, 1.4-, and 1.2-times strain to failure compared to unfilled PDMS, respectively.

#### 3.2.2. Tear Strength

Plots of tear strength versus weight percent are presented in Figure 11. Like tensile strength, tear strength for the three filler types increased in a non-linear manner as filler content increased. No significant differences in tear strength were noted at lower filler loadings (0%, 0.5%, and 5%, *p* < 0.05) for the three types of filled materials. Tear strength of KH220 was significantly greater than uncoated and TS530 at 10% and 15% loadings, while tear strength of 10% filled TS530 was greater than uncoated at 10% loadings (*p* < 0.05), but not significantly different at 15% load levels (*p* ≥ 0.05). The highest tensile strength was recorded for 15% filled KH220 materials, which were 1.2 and 1.3 times higher than 15% uncoated- and TS530-filled materials, respectively. Compared to unfilled PDMS, 15% KH220-, uncoated- and TS530-filled materials were 5.8, 4.8, and 4.4 times stronger, respectively.

#### 3.2.3. Shore a Durometer Hardness

Figure 12 shows hardness differences among indoor groups before and after 3000 h. As observed for other properties, hardness increased non-linearly with increasing filler levels for the three filler types. For indoor storage there were no differences in hardness before and after 3000 h for all filler loadings and filler types, except for 15% loaded TS530 materials, which increased approximately 3 Shore A units (*p* < 0.05). Hardness results for outdoor weathering appear in Figure 13, and similar patterns with indoor groups were observed. No differences in hardness before and after weathering were noted, except for 10% and 15% loaded TS530 elastomers (*p* < 0.05). These represented 2.6 and 2.4 hardness unit increases, respectively. The highest durometer hardness occurred with 15% KH220-filled materials, which were 1.2 times harder than 15% TS530-filled materials, but not significantly different from 15% uncoated-filled materials.

#### 3.2.4. Translucency Parameter

Translucency parameter was calculated from L* a* b* differences measured on black and white backgrounds. Figure 14 displays translucency parameter values initially and following 3000 h of indoor storage in darkness. No differences in translucency parameter were noted between before and after 3000 h, regardless of filler loading or type (*p* ≥ 0.05). There was a trend of decreasing translucency parameter values with each increase in filler loading for KH220- and uncoated-filled materials, but TS530 initially decreased, then remained unchanged at filler loadings above five weight percent.

Effects of outdoor weathering are shown in Figure 15. All filler types experienced significant decreases in translucency parameter for 0% and 0.5% filler content, which ranged from 50.91 to 45.03 and 44.96 to 39.24, respectively (*p* < 0.05). Highly filled materials (10%, 15%) maintained translucency after outdoor weathering, with nanofilled materials possessing translucency parameter values 18 to 20 units lower than TS530-containing materials. From visual observation of the samples, lower filler loadings produced color changes that were darker and more opaque following 3000 h of outdoor weathering, whereas undetectable color changes were observed in higher filled samples.

#### 3.2.5. Viscoelastic Measurement

Figure 16, Figure 17, Figure 18 and Figure 19 show storage modulus (E′), loss modulus (E″) and tan delta curves from unfilled, 15% coated nanosilica-, 15% uncoated nanosilica-, and 15% submicron silica-filled PDMS, respectively. A single peak was observed in E″ and in tan delta, along with a drop of E’, in all samples in the temperature region of the first glass transition, occurring between −110 to −95 °C (Transition I). At higher temperatures, −95 to −40 °C, E′ and E″ decreased gradually, with E′ in 15% coated nanosilica-filled PDMS observed to have steeper curves compared to other groups. Additional tan delta peaks were observed at temperatures near −70 °C (Transition II) and −40 °C (Transition III). A rubbery plateau region was observed between −40 °C and +20 °C for all samples.

Filler type and weight percent filler had a significant effect with storage modulus in Transition I and weight percent filler had a significant effect in tan delta at Transition II. Otherwise, filler type and weight percent filler exerted little effect on any of the viscoelastic properties at the three transition temperatures (*p* ≥ 0.05). In the rubbery plateau region near room temperature (around 20 °C), both filler type and weight percent filler exerted significant differences on storage and loss moduli for the three filler types (*p* < 0.05). The highest storage modulus, loss modulus, and tan delta values were recorded for the two 15%-nanoparticle-filled materials, approximately 1.1–1.4 MPa higher than unfilled and lowly filled PDMS. Storage modulus decreased after Transition I for all filler types.

Table 2 presents pairwise comparison results for primary glass temperatures occurring at Transition I, II, and III. There was a significant effect of filler type and weight percent filler on Transition I and Transition II temperatures (*p* < 0.05). No significant differences were noted for Transition III temperatures (*p* ≥ 0.05). All levels of filled PDMS demonstrated higher Transition I temperatures compared to unfilled PDMS. Transition I temperatures at all filler loadings were not significantly different between KH220- and TS530-filled materials, whereas KH220 was significantly higher than uncoated-filled materials at 5% and 10% filler loadings.

## 4. Discussion

### 4.1. Tensile and Tear Properties

Compared to uncoated nano-SiO_2_ or conventional submicron SiO_2_, ultimate tensile strength and elastic modulus of superhydrophobic-coated nano-SiO_2_ filled samples were greatest at 15% filler loading (roughly 1.3–1.4 times greater tensile strength, 1.5–2.5 times greater elastic modulus, Figure 4 and Figure 5). This was primarily attributed to high filler loadings and to properties attributed to nanoparticles. These include an increased number of particles per unit mass loaded into the polymer, increased surface reactivity and improved dispersion when coated with a hydrophobic compound. It is well documented that increasing filler levels increase physical properties in polymer systems. In dental resin-based composites, properties such as compressive and tensile strengths, elastic modulus, hardness and wear resistance are increased with sequential additions of micrometer and submicrometer size fillers. Water sorption and hydrolytic degradation are decreased [23,24]. Similar results have been shown for filled PDMS systems, where tensile strength, tear strength, and hardness are increase with the addition of nanometer size fillers [14,25].

In this study, tensile strength was 420% to 600% higher for the three 15% filled materials compared to unfilled PDMS. In unfilled polymer, applied tensile stress serves to straighten the polymer chains, and microvoids are formed as means to redistribute molecular voids in the polymer. These microvoids continue to grow into failure since voids act as stress concentrators which trigger the onset of failure at lower stresses [26]. In filled systems, silica fillers serve to reinforce the polymer by filling the voids. Due to reinforcement generated by silica’s high surface area and the formation of hydrogen bonds between the filler’s hydroxyl groups and Si and CH_2_ in the siloxane polymer, the properties are significantly improved [27]. This permits adsorption of filler to polymer molecules, better reinforcement under load, higher tensile strength, and elongation capability [28]. In addition, reinforcement of PDMS is also contributed by particle-particle interaction, where the resistance to the applied force is substantially increased and shrinkage during polymerization is minimized by hydrogen bonding between free hydroxyl groups on surface particles [29].

As mentioned, the ten-fold size reduction from submicron- to nano-sized particles increases surface area by approximately 1000 and decreases volume by approximately 1000, producing a surface area/volume ratio of 0.15, which is ten times higher. Assuming the same mass is present within materials loaded with the different particles, this roughly equates to ten times more nanoparticles per unit volume. The surface area/weight ratio increases by one to two orders of magnitude as the nanoparticles decrease in size from 100 to 10 nm [30] The increased particle numbers, in conjunction with a higher total surface area, provides more reinforcement sites with a more efficient stress transfer mechanism, which in turn increases strength [31] The confinement of filler networks further contributes to better crystallization because it generates nucleation sites that hasten the crystallization process in addition to the interactions between the polymer and filler [32,33]. From this information, it is reasoned that an increase of crystallization produced crystallites that restricted chain movement, which produced higher strength and modulus. This underlies the differences observed in tensile strength and elastic modulus for 15%-filled PDMS compared to lower weight percentages.

Untreated silica fillers have moisture-attracting silanol groups (SiOH) on the surface which are formed as a result of rehydroxylation of silica when exposed to water or aqueous solutions. Even the presence of humidity readily populates the surfaces with hydroxyl groups. The presence of silanol groups increase electronegativity and renders a hydrophilic surface. The hydroxyl groups (OH) act as the centers of molecular adsorption and capable of forming a hydrogen bond with other OH groups [34]. This hydrogen bonding results in strong filler-filler attraction and tight aggregates can form, causing poor dispersion of silica into the polymer [35]. Hydrophobic coatings decrease the number of sites where silanol groups reside on the silica surface of silica, as silane coatings react with or displace the silanol groups. The net result is to reduce particle-particle attraction and hence agglomeration, which in turn promotes improved dispersion of filler into the uncured siloxane oligomer [36,37,38]. Unfortunately, when manufactured and stored as powders, the silica particles can agglomerate through condensation reactions at interparticle contacts that are generated during the drying process, which renders dispersibility more challenging [36].

Submicron SiO_2_-filled samples produced significantly greater failure strain compared to both nano-SiO_2_ filled materials at all filler loadings. At 15% filler, failure strain was 1.4–1.6 times higher for submicron-filled siloxanes. Because elastomers undergo little, if any, plastic deformation prior to failure, failure strain changes inversely with the modulus. That is, stiffer materials undergo less strain at failure. An increased number of fillers and improved polymer adsorption by nanosized particles increase adhesion between particle-polymer and enhancement polymer stiffness. This, in turn, restricts polymer chain movement and hence elongation at tensile failure. Similar observations have been reported when nanosilica are added to commercial silicone elastomers containing submicron particles. Additions above 1.5 weight percent decrease tensile failure strain [14].

Similar to tensile strength, superhydrophobic- coated nano-SiO_2_-filled PDMS produced higher tear strength compared to uncoated nano-SiO_2_- and submicron-filled PDMS. The tear strengths of coated nano-SiO_2_ materials were approximately 1.2 times higher than two other filler types at 15% filler loading. Compared to unfilled PDMS materials, this represented a 640% increase. Tear strength represents resistance to crack propagation, and in composite materials, additional energy is needed to propagate a crack through a stiffer material, and/or grow a crack either around or through the particles. The excellent tear strength observed for coated nano-SiO_2_-filled PDMS materials was attributed to strong adhesion between nanoparticles and polymer matrix, which effectively stiffened the materials and offered a physical barrier against growing cracks [39]. The stiff particles restrict crack formation, hence more energy is needed for crack propagation. The surface area of the expanding fracture and the energy required to propagate it increase as a result of nanoparticles diverting the crack off its primary plane [40].

Similar findings have been reported by Zayed et al., where the addition of 3% surface treated SiO_2_ nanofiller to a commercial silicone produced significant improvements in tear strength compared to materials without nanoparticles. Chemical treatment of a silica filler with silane has been shown to improve filler incorporation into and reinforcement of the siloxane, resulting in increased material strength and tear resistance [41].

### 4.2. Durometer Hardness

The higher hardness values observed in nano-SiO2-filled polymers is attributed to the reinforcing behavior of nanoparticles, which increases stiffness and is expected to reduce the viscous response. Consequently, higher hardness is rendered, as shown in Figure 8 and Figure 9. Durometer hardness was not significantly affected by 3000 h of outdoor weathering in both nano-SiO_2_ filled groups. Meanwhile, after weathering, statistically significant hardness increases were observed in 10% and 15% submicron-filled polymers. Ultraviolet radiation degrades polymers through a radiolysis mechanism which enhances cross-linking and the production of smaller polymer chains that leads to volatile degradation products [42,43,44]. With ultraviolet radiation exposure being the primary cause of weathering-induced damage, the ability of fillers to block incident radiation is key to protecting the surrounding polymer from chemical change. With as much as 10 times more nanoparticles present per unit volume, enhanced protection is offered over submicron particles, and chemical changes to the polymer are reduced. However, it should be acknowledged that the 2.6-unit maximum increase in Shore-A hardness observed by submicron-containing materials may not be clinically detectable when touching the surface of a facial prosthesis. This implies that clinically, these materials are equally adequate in maintaining hardness in the face of outdoor weathering. In previous studies, there were progressive hardenings of commercial prosthetic silicone elastomers after 9 months of outdoor weathering and hardening of a medical-grade PDMS after one-year outdoor weathering [44]. It is noteworthy that materials tested in these studies possessed submicron silica, which accounts for TS530-filled materials mirroring their results.

### 4.3. Translucency Parameter

A material’s translucency is dependent upon the combined effects of absorption and scattering. Light transmission through a material is affected by its composition, and for filled PDMS, both polymer matrix and fillers control the amount of light scattering and absorption. Filler volume fraction, filler size, and refractive indices of polymer and fillers affect light scattering and absorption [45]. In this study, both coated and uncoated nano-SiO_2_-filled elastomers produced lower translucency parameter values than submicron SiO_2_-filled elastomers. Because nanoparticles provide large total surface area, they act to better block the reflectance off white and black background to incident light. Therefore, their lower translucency parameters indicate greater masking abilities of underlying structures, which reduces the need for opacifiers added to a prosthesis [21]. This permits more accurate formulation of skin tones, as the whitening effect of opacifiers is reduced or eliminated.

For the two nanoparticle types, unfilled and lowly filled polymers (0%, 0.5%, 5%) experienced a five unit decrease in translucency parameter values after 3000 h of outdoor weathering. This indicated the materials became more opaque. In visual observation, the samples darkened. Colorimetric changes demonstrated decreases of L* values (approximately 1 to 4 units) and increases of a* and b* values (approximately 0.1 to 2 units), which meant the materials primarily blackened and lightly turned more blue and red. Based on limits of perceptibility (1.3 units) and acceptability (4.4 units) in translucency parameter established by Paravina et al., the translucency changes (5 units) were not acceptable and would produce a mismatch in appearance between prosthesis and skin [46]. This implies that a minimum of 10% nanofiller is needed to prevent these changes. These results are consistent with those reported by Tukmachi et al., where a significant decrease of light transmission through high-nanosilica-filled PDMS occurred, compared to lower percentages nanosilica [25]. In dental resin composite, Lee et al. reported that composite filled with nanoparticles with higher filler content produced lower translucency parameter values [47].

### 4.4. Dynamic Mechanical Analysis (DMA)

DMA is a useful technique for experimental characterization of viscoelastic properties of polymers. DMA measures the dynamic modulus and viscoelastic damping under dynamic vibrational loading at different temperatures or frequencies. These properties change significantly when segmental mobility increases, and crystalline structures transition to the amorphous phase. Temperature dependence is best identified when the test is conducted in stress-temperature mode, which was chosen for this study.

At −150 °C PDMS behaves as a glass, where polymer molecules are well anchored by hydrogen bonding and low thermal disturbance. This produces high values for storage modulus and low values for loss modulus, rendering the deformation response as elastic in nature. Temperature increases to Transition I decrease storage moduli and increase loss moduli for all formulations, including unfilled (Figure 16, Figure 17, Figure 18 and Figure 19). These moduli changes reflect a decrease in stiffness and increase in damping, which increases tan delta and appears as a spike on the graphs. At this temperature, the polymer is undergoing devitrification and reforming crystallites that were generated during cooling to −150 °C. A comparison of Transition I temperatures presented in Figure 16, Figure 17, Figure 18 and Figure 19 and Table 2 demonstrate approximately 4 °C higher transition temperatures for 15%-filled formulations compared to unfilled PDMS. Since increasing temperatures are causing polymer chains to fold and align into crystallites, materials with more crystals require more thermal energy and hence higher temperatures. This also can be understood by viewing the reverse process, where materials with higher filler numbers more effectively nucleate crystals during cooling to −150 °C. Transition I temperatures occur earlier, at higher temperatures. Temperature transition trends appearing in Table 3 also show for 5% and higher filled materials that KH220-filled PDMS consistently demonstrates higher transition temperatures than do uncoated- and TS530-filled PDMS. This is consistent with higher crystal numbers created by superhydrophobically coated particles promoting superior filler dispersion, where more nucleation sites are available.

A second tan delta peak is detected in curves for all samples at Transition II. During cooling from the melt, the PDMS molecules begin to undergo crystallization and form semi-crystalline polymers. Additional topological restrictions are imposed by crystallites, which decrease the segmental mobility of polymer chains and cause both storage and loss moduli to rise as the polymer cools. This can be seen as increases in slopes of moduli curves in Figure 16, Figure 17, Figure 18 and Figure 19. As crystallization proceeds with lowering temperatures, the polymer slowly stiffens. The presence of silica fillers provides nucleation sites for crystallization, thereby accelerating the crystallization process [33,48]. At Transition II, filler type and weight percent filler significantly affect transition temperatures. Temperatures presented in Table 3 demonstrate lower temperatures for 5% and 10% TS530-filled PDMS compared to coated- and uncoated-filled PDMS. Because filler presence promotes crystallization, more fillers equate to more crystals and higher crystallization rates. With ten times more particles present in nanofilled formulations, crystallization occurs sooner during cooling and hence produces higher Transition II temperatures.

At transition III, a high spike of tan delta occurs, as shown in Figure 16, Figure 18 and Figure 19, indicating viscous behavior dramatically increases, thereby raising E″/E′ ratio. Transition III corresponds to melt temperature during heating and solidification during cooling. For elastomers, in-service stretchability at ambient temperature is referenced as the melt condition. At this temperature, a partially crystalline solid is converted to a rubbery one during heating through “melting” of the crystallites, causing remarkable damping to occur. Careful inspection of Figure 16, Figure 18, and Figure 19 reveal an increase in storage modulus that produces a small “hump” in the curve at temperatures below the melt temperature. During cooling, the loss of rubbery behavior occurs through the start of crystal formation. Consequently, polymer stiffening occurs and produces the hump. For uncoated and TS530-filled materials, Transition III produces trends similar to those observed for Transitions I and II, where increasing filler content tends to drive the transition temperature upward (Table 3). However, KH220-filled materials show erratic raising and lowering of temperature when filler increases, which are unexplainable. Regardless of temperature differences, physical properties are unaffected, as storage and loss moduli are similar for all materials at all filler loadings at the Transition III temperature.

As temperatures increase into the plateau region during heating, storage moduli values for all formulations decrease approximately ten-fold as 20 °C is approached. Higher storage moduli values occur as filler levels increase for the three filler types, with highest storage moduli present for nanosilica-filled formulations, for reasons previously discussed. Decreases in loss moduli over this temperature range are more dramatic, as 100-fold decreases place loss moduli values in the 1–20 kPa range. Overall, this produces the highly elastic material response characteristic of elastomers at ambient temperatures.

## 5. Conclusions

PDMS elastomers loaded with 15% superhydrophobic coated nano-SiO_2_ produced the highest tensile strength, elastic modulus, tear strength and durometer hardness. PDMS elastomers filled with 15% submicron SiO_2_ produced the highest failure strain and translucency parameter values. Durometer hardness was not affected by 3000 h outdoor weathering in all groups, except for 10%- and 15%-filled submicron SiO_2_ elastomers, which hardened by 2.6 hardness units. Unfilled and 0.5% filled elastomer experienced decreases of translucency parameter values after 3000 h of outdoor weathering, while higher filled loading materials maintained their translucency parameter values. Transition temperatures of filled PDMS were higher than those for unfilled PDMS. It was determined that superhydrophobic-coated nano-SiO2-filled PDMS created the strongest, most tear-resistant, and least translucent materials which improved the physical properties of maxillofacial prosthesis. However, it also produced materials with lower stretchability and higher hardness, which were viewed as drawbacks for producing a “skin-like” feel.

## Figures and Tables

**Figure 1 materials-15-07343-f001:**
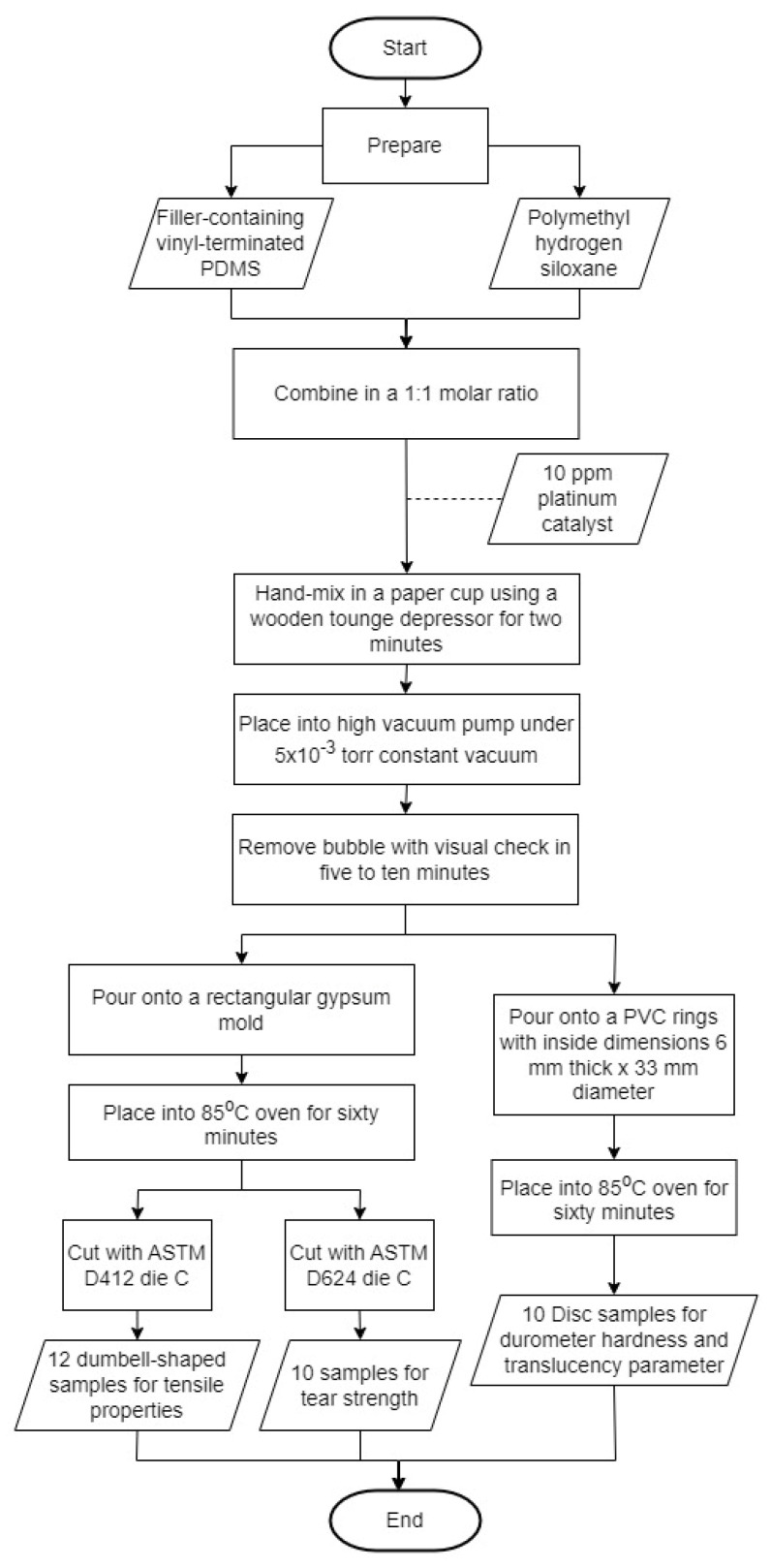
Sample preparation flowchart.

**Figure 2 materials-15-07343-f002:**
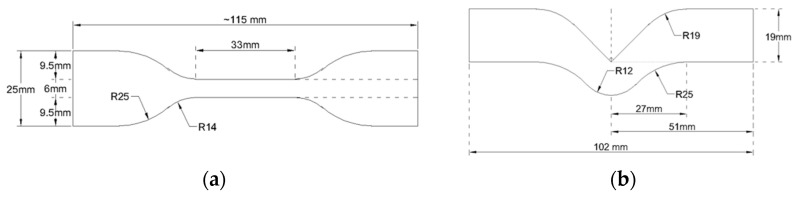
(**a**) ASTM D412 die C sample dimension for tensile properties measurements. (**b**) ASTM D624 die C sample dimension for tear strength measurement.

**Figure 3 materials-15-07343-f003:**
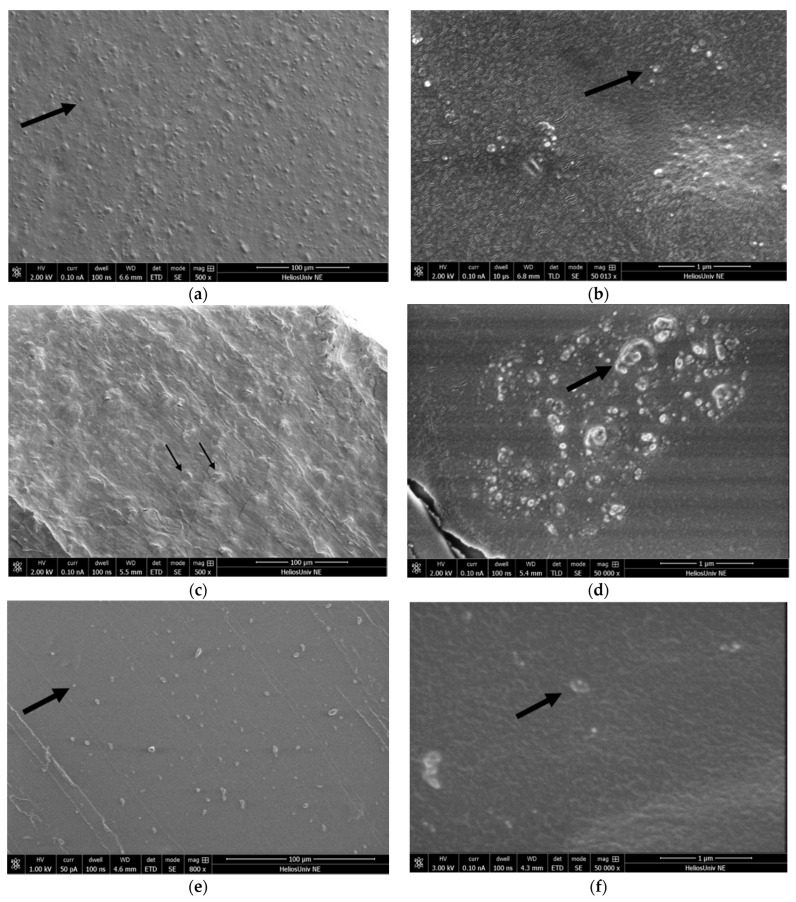
(**a**) Dispersion of 15% KH220 filler loading at 500× magnification. Nanoparticles appear to be evenly distributed into the polymer. Arrow indicates a nanoparticle. (**b**) Dispersion of 15% KH220 filler loading at 50,000× magnification. Arrow points to a nanofiller particle approximately 20 nm diameter. (**c**) Dispersion of 15% Uncoated Filler loading at 500× magnification. Arrows indicate agglomeration of nanoparticles. A large number of aggregates can be seen throughout the polymer. (**d**) Dispersion of 15% Uncoated Filler loading at 50,000× magnification. Image demonstrates up to 0.4 μm diameter agglomerated nanoparticles (arrow), which are approximately 20 times larger than nanoparticles observed in Figure 2b. (**e**) Dispersion of 15% TS530 filler loading at 800× magnification. Particles appear to be evenly distributed into the polymer, however, fewer particles are observed compared to nanofillers with similar magnifications. Arrow indicates submicron particle. (**f**) Dispersion of 15% TS530 filler loading at 50,000× magnification. Arrow points to filler particle approximately 200 nm in diameter.

**Figure 4 materials-15-07343-f004:**
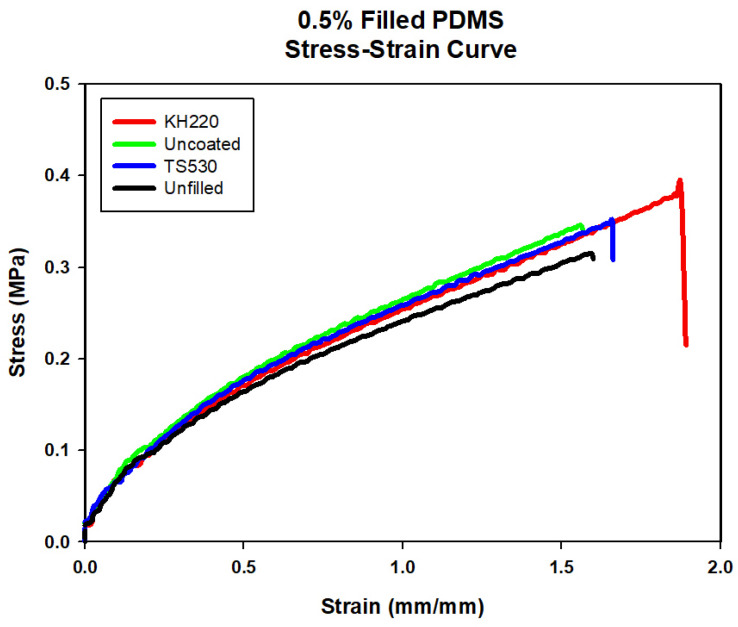
Stress–strain Curve of Unfilled and 0.5% Filled PDMS with Three Filler Types.

**Figure 5 materials-15-07343-f005:**
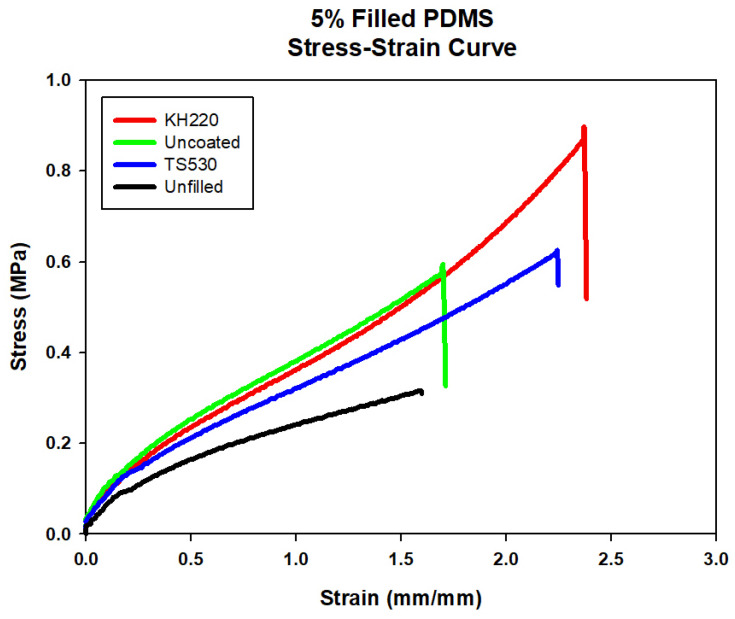
Stress–strain Curve of Unfilled and 5% Filled PDMS with Three Filler Types.

**Figure 6 materials-15-07343-f006:**
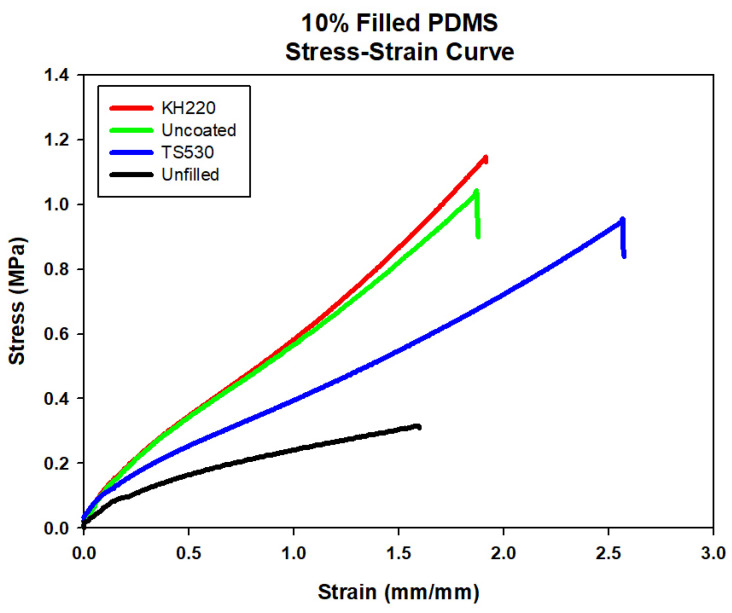
Stress–strain Curve of Unfilled and 10% Filled PDMS with Three Filler Types.

**Figure 7 materials-15-07343-f007:**
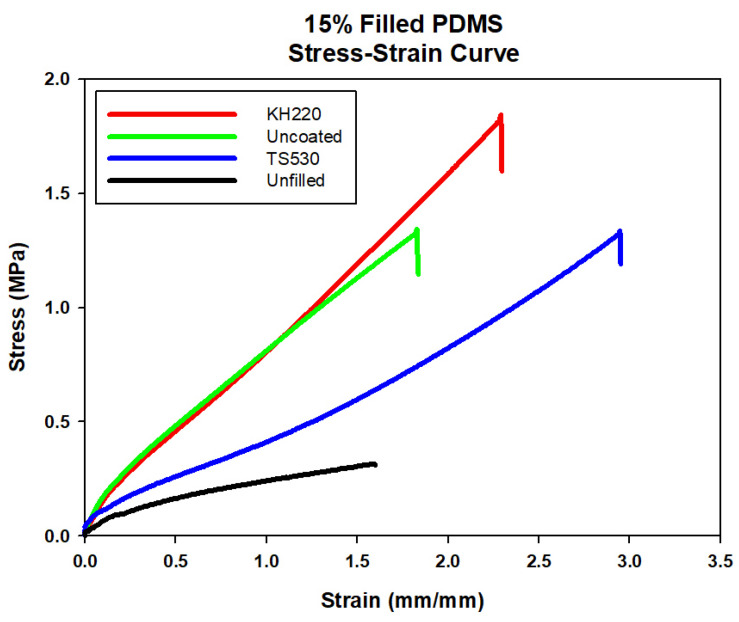
Stress–strain Curve of Unfilled and 15% Filled PDMS with Three Filler Types.

**Figure 8 materials-15-07343-f008:**
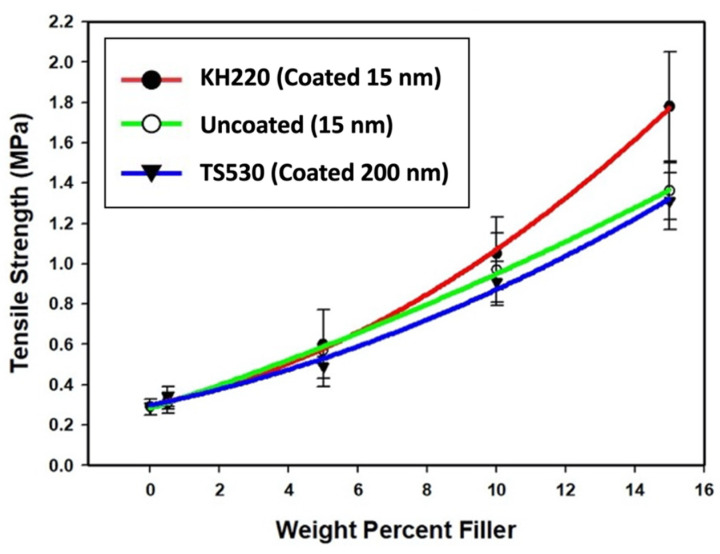
Plots of Tensile Strength (MPa) Versus Weight Percent Filler For Three Filler Types. Error Bars Represent Standard Errors of Means.

**Figure 9 materials-15-07343-f009:**
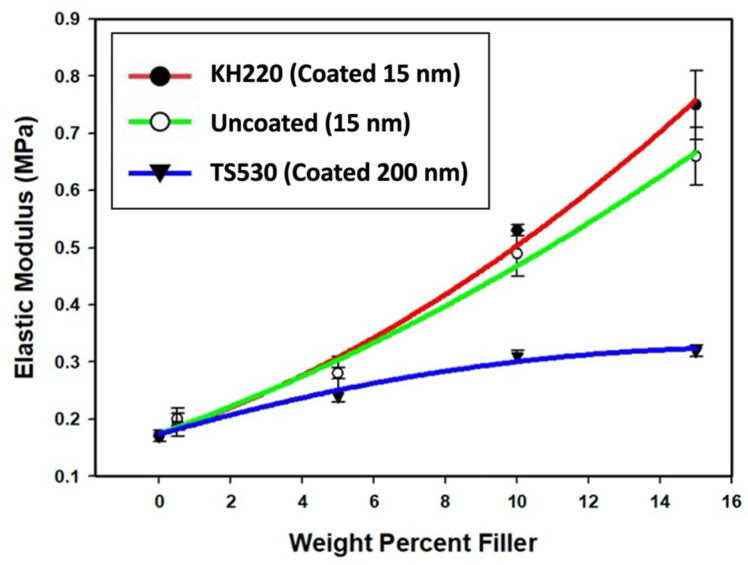
Plots of Mean Elastic Modulus (MPa) Versus Weight Percent Filler For Three Filler Types. Error Bars Represent Standard Errors of Means.

**Figure 10 materials-15-07343-f010:**
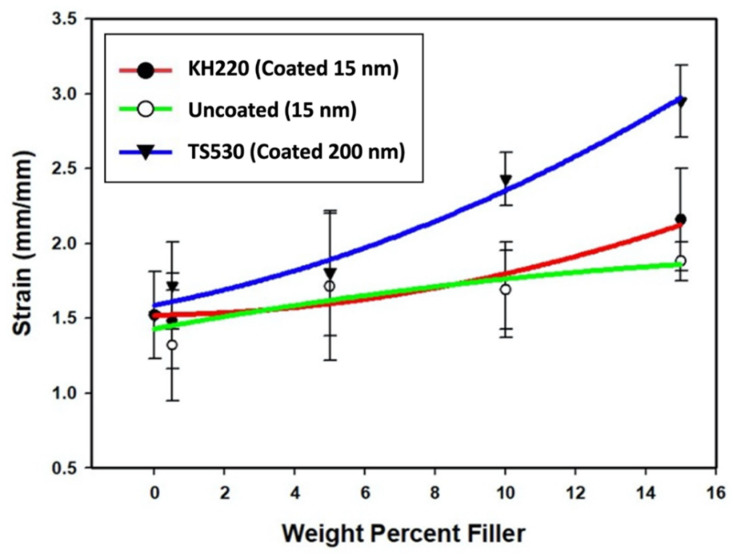
Plots of Failure Strain (mm/mm) Versus Weight Percent Filler For Three Filler Types. Error Bars Represent Standard Errors of Means.

**Figure 11 materials-15-07343-f011:**
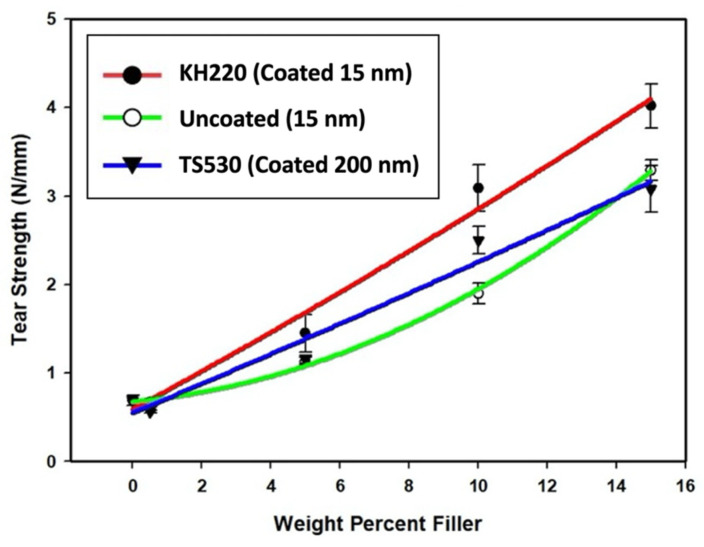
Plots of Tear Strength (N/mm) Versus Weight Percent Filler For Three Filler Types. Error Bars Represent Standard Errors of Means.

**Figure 12 materials-15-07343-f012:**
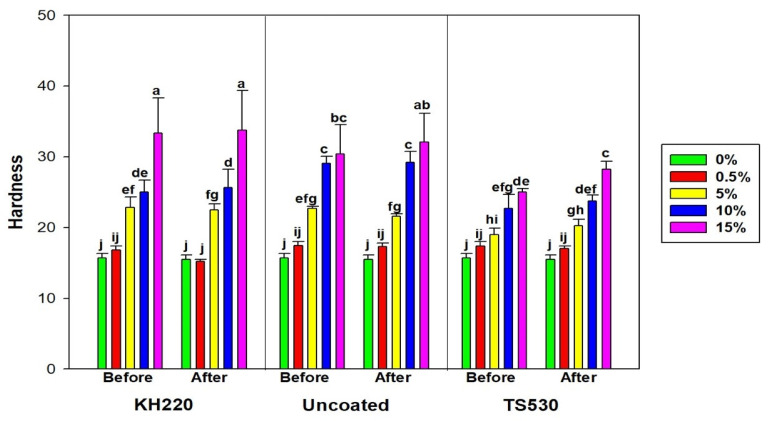
Bar graphs displaying means and error bars representing standard errors of means for Durometer Hardness for Indoor samples before and after 3000 h of storage. Lower case letters denote statistical groupings. Groups with means having the same lower-case letters are not significantly different (*p* > 0.05, ANOVA/Tukey). Comparisons are both within and across materials.

**Figure 13 materials-15-07343-f013:**
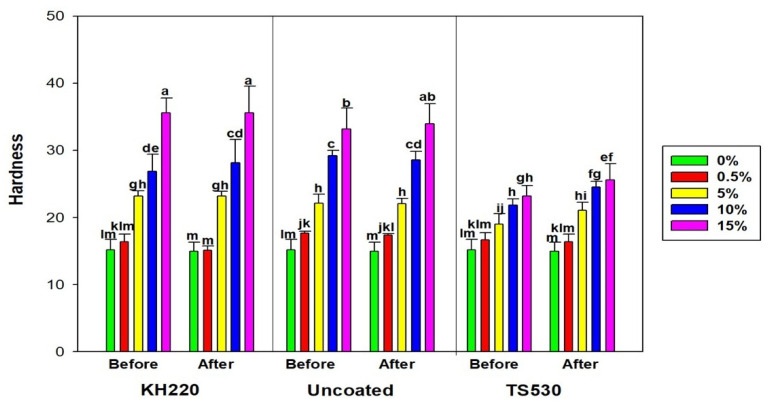
Bar graph displaying means and error bars representing standard errors of means for Durometer Hardness for Outdoor samples before and after 3000 h of weathering. Lower case letters denote statistical groupings. Groups with means having the same lower-case letters are not significantly different (*p* > 0.05, ANOVA/Tukey). Comparisons are both within and across materials.

**Figure 14 materials-15-07343-f014:**
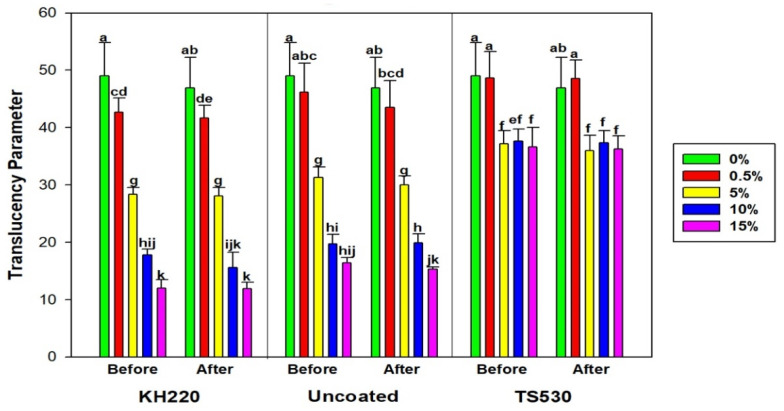
Bar graph displaying means and error bars representing standard error of Translucency Parameter for Indoor samples before and after 3000 h of storage. Lower case letters denote statistical groupings. Groups with means having the same lower-case letters are not significantly different (*p* > 0.05, ANOVA/Tukey). Comparisons are both within and across materials.

**Figure 15 materials-15-07343-f015:**
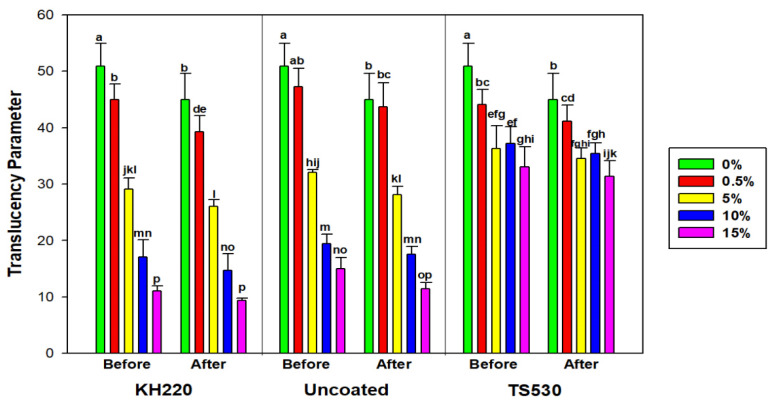
Bar graph displaying means and error bars representing standard errors of means for Translucency Parameter for Outdoor samples before and after 3000 h of weathering. Lower case letters denote statistical groupings. Groups with means having the same lower-case letters are not significantly different (*p* > 0.05, ANOVA/Tukey). Comparisons are both within and across materials.

**Figure 16 materials-15-07343-f016:**
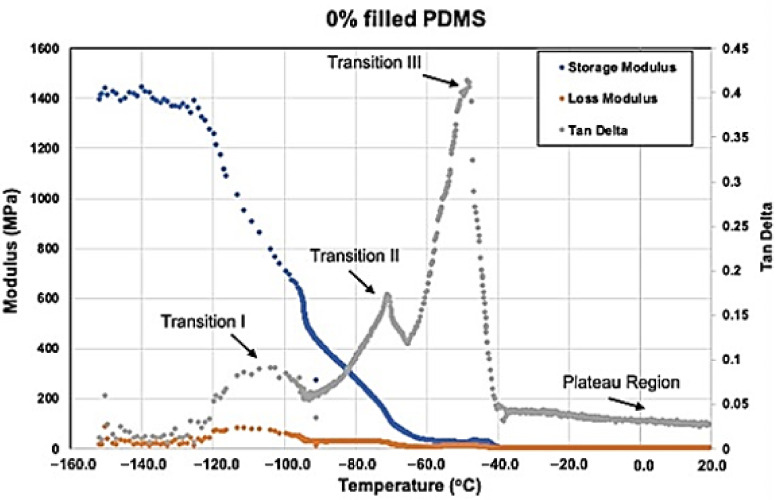
Storage Modulus, Loss Modulus, And Tan Delta Curves For 0% Filled PDMS From −150 °C To +20 °C. (Transition I not labeled).

**Figure 17 materials-15-07343-f017:**
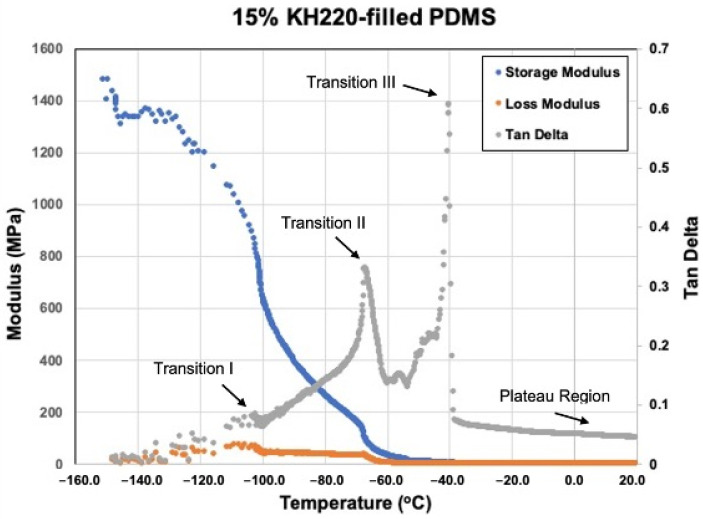
Storage Modulus, Loss Modulus, And Tan Delta Curves For 15% KH220-filled PDMS From −150 °C To +20 °C.

**Figure 18 materials-15-07343-f018:**
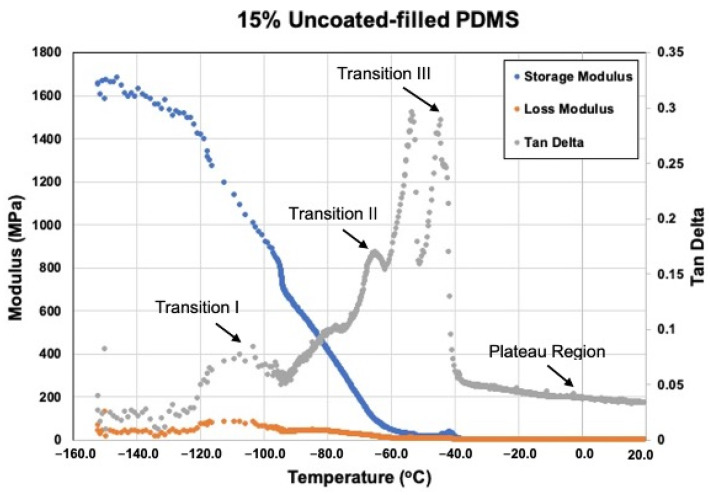
Storage Modulus, Loss Modulus, And Tan Delta Curves For 15% Uncoated-filled PDMS From −150 °C To +20 °C.

**Figure 19 materials-15-07343-f019:**
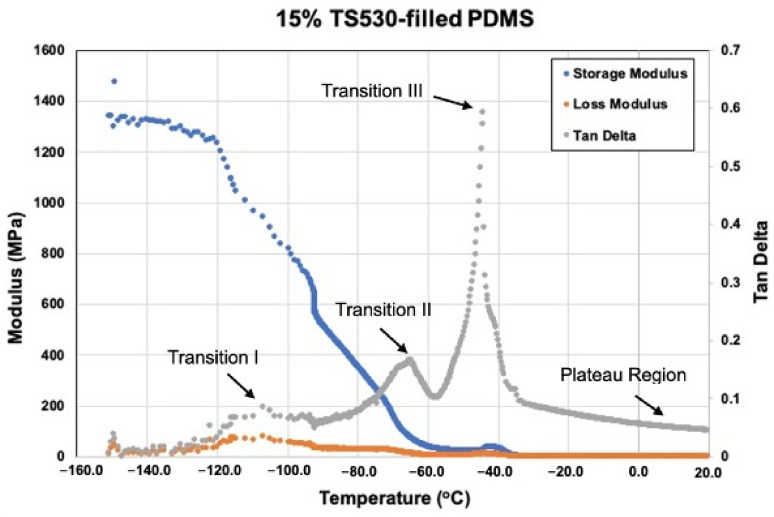
Storage Modulus, Loss Modulus, And Tan Delta Curves For 15% TS530-filled PDMS From −150 °C To +20 °C.

**Table 1 materials-15-07343-t001:** Materials and Manufacturers.

Materials	Manufacturers
V2K Vinyl-Terminated PDMS	Momentive Materials, Tarrytown, NY, USA
15 nm Dimethoxydiphenylsilane-coated nano-SiO_2_ (KH220)	US Research Nanomaterials, Inc., Houston, TX, USA
15 nm Uncoated nano-SiO_2_	US Research Nanomaterials, Inc., Houston, TX, USA
200 nm Hexamethyldisilazane-coated SiO_2_ (TS530)	Cabot Corporation, Boston, MA, USA
V-XL crosslinker	Momentive Materials, Tarrytown, NY, USA
10 ppm Platinum catalyst	Momentive Materials, Tarrytown, NY, USA

**Table 2 materials-15-07343-t002:** *p*-Values From Pairwise Comparisons For Transition I, II, and III Temperatures. Results From ANOVA/Tukey.

**Two-Way ANOVA of Transition I**					
**Source**	**DF**	**Sum of Square**	**Mean Square**	**F Value**	**Pr > F**
Filler Type	2	15.25911111	7.62955556	4.21	0.0245
Weight Percent	4	47.53688889	11.88422222	6.55	0.0006
Weight Percent × Filler Type	8	14.46977778	1.80872222	1.00	0.4582
**Two-Way ANOVA of Transition II**					
**Source**	**DF**	**Sum of Square**	**Mean Square**	**F Value**	**Pr > F**
Filler Type	2	274.3093333	137.1546667	3.35	0.0487
Weight Percent	4	649.1475556	162.2868889	3.96	0.0107
Weight Percent × Filler Type	8	386.3617778	48.2952222	1.18	0.3438
**Two-Way ANOVA of Transition III**					
**Source**	**DF**	**Sum of Square**	**Mean Square**	**F Value**	**Pr > F**
Filler Type	2	299.377333	149.688667	0.79	0.4614
Weight Percent	4	1196.134667	299.033667	1.59	0.2036
Weight Percent × Filler Type	8	1492.509333	186.563667	0.99	0.4637

**Table 3 materials-15-07343-t003:** Mean ± SE of Transition I, II, and III Temperatures. Results From ANOVA/Tukey *.

Filler Type	Weight Percent	Transition I (°C) Mean ± SE	Transition II (°C) Mean ± SE	Transition III (°C) Mean ± SE
	0	−103.3 ± 0.64 ^D^	−67.6 ± 3.57 ^A^	−47.0 ± 4.78
KH220	0.5	−101.3 ± 1.16 ^ABC^	−68.5 ± 5.02 ^A^	−44.6 ± 3.65
	5	−99.8 ± 0.66 ^A^	−71.1 ± 1.01 ^A^	−47.6 ± 1.53
	10	−99.7 ± 1.39 ^A^	−73.2 ± 8.04 ^AB^	−49.5 ± 3.53
	15	−99.8 ± 1.31 ^A^	−65.8 ± 5.91 ^A^	−43.7 ± 2.96
Uncoated	0.5	−101.4 ± 0.67 ^ABCD^	−67.3 ± 1.27 ^A^	−47.3 ± 1.83
	5	−102.3 ± 0.34 ^BCD^	−68.7 ± 4.01 ^A^	−47.5 ± 2.33
	10	−102.6 ± 1.93 ^CD^	−75.2 ± 14.51 ^ABC^	−46.7 ± 4.84
	15	−101.1 ± 2.20 ^ABC^	−71.7 ± 8.96 ^AB^	−44.2 ± 0.75
TS530	0.5	−100.0 ± 2.07 ^A^	−67.3 ± 4.84 ^A^	−48.1 ± 5.39
	5	−101.3 ± 0.87 ^ABCD^	−84.6 ± 3.50 ^C^	−44.2 ± 3.50
	10	−100.5 ± 1.53 ^ABC^	−82.4 ± 6.86 ^BC^	−44.3 ± 2.07
	15	−100.1 ± 1.59 ^AB^	−65.8 ± 7.63 ^AB^	−43.4 ± 5.23

* Means with the same superscript letter are not significantly different (*p* ≥ 0.05). Vertical comparisons only.

## Data Availability

The data presented in this study are available on request from the corresponding author.

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
