# Peer review of "Effect of Superhydrophobic Coating and Nanofiller Loading on Facial Elastomer Physical Properties"

_materials, 2022, doi:10.3390/ma15207343_

Round 1
Reviewer 1 Report
This work showed that the inclusion of 15% superhydrophobic coated nano-SiO2 into a PDMS matrix increases mechanical properties, in particular, the tensile strength, elastic modulus, tear strength and durometer hardness.
My suggestions for the authors are:
Line 76: remove the following text " purpose of this thesis project"
Add units of stress, strain, cross-section (Equation 1)
Merge Figure 1,2,3 into just one Figure.
What is the meaning of lowercase letters in figure 8,9. this strategy to indicate statistical differences is not very readers friendly.
It would be very interisting provide images of each material before and after UV exposure.
Reviewer 2 Report
This manuscript introduced PDMS/SiO2 nanocomposites and used them as artificial skin. The mechanical properties were systematically evaluated. I recommend publication after the following questions are addressed.
1. What is the rationale behind the design of polymer? Why did the authors used "1: 1 molar ratio with polymethyl hydrogen siloxane", which is quite a lot for vinyl terminated PDMS. Usually, 5 parts of PMHS is enough for 100 parts of vinyl terminated PDMS. Please explain.
2. Did the authors take SEI pictures of the surface of the samples to evaluate the dispersion of nano-fillers? Fracture surfaces are more convincing.
3. Please describe the methods in detail. For example, the sputtering process in SEM method is missing.
4. Please insert stress-strain curves of the samples.
Reviewer 3 Report
The paper is quite lengthy, however, it presents very interesting data regarding the effect of hydrophobic-coated nano-SiO2 on the physico-mechanical properties of polydimethylsiloxane.
I have a few comments/suggestions for authors as follows:
1. Title needs to be restructured so that it reflects the objective of study.
2. Authors have used the words 'mechanical' and 'physical' interchangeably, it is suggested that authors should maintain the uniformity in order to avoid confusion.
3. Authors have evaluated the viscoelastic properties using 'Dynamic Mechanical Analyzer' then why did the authors determine the separate static elastic modulus? Please justify
4. It is suggested that the authors should merge all tables (2 to 7) if possible.
5. The size of labels in figures 12, 13, 14 and 15 should be increased.
6. Authors may add the clinical significance in the conclusion on the basis of current findings of this study.
7. Lastly, it suggested that authors should briefly add a paragraph in the introduction section regarding the relevance of properties they have evaluated in their work.
Reviewer 4 Report
This paper investigated the effect of nanofiller coating and loading on facial elastomer physical properties. The coated and uncoated samples were compared by a series of tests. This paper can be improved by addressing the following issues:
1. In the last paragraph of the introduction part, the work in this paper should be described with more details.
2. In the sample preparation, a flow chart will make it much clearer because there are multiple steps in the preparation process.
3. The shape and dimensions of the sample should be shown clearly in a figure.
4. In section 2.2.3, more details about the hardness test should be provided. There are many kinds of hardness tests.
5. In the results, the original stress/strain or load/displacement curve should be provided for better understanding the mechanical behavior of the sample.
6. The conclusion should be written in paragraph, rather some bullet points.
